cruise(s); blue economy; sustainability; economic impact(s); systematic literature review

**Author for correspondence**:
Alexis Papathanassis,
Email: apapathanassis@hs-bremerhaven.de

# A decade of 'blue tourism' sustainability research: Exploring the impact of cruise tourism on coastal areas

Alexis Papathanassis [ID]

Bremerhaven University of Applied Sciences, Faculty of Management and Information Systems, Cruise Tourism Management, Bremerhaven, Germany

## Abstract

Cruise tourism research has developed exponentially during the past decades. Global tourism activity in general and cruises in particular are concentrated in coastal areas and represent a dominant part of the so-called 'blue economy'. Within this context, the public debate surrounding the impact of cruise tourism on port communities reflects a narrative of unsustainable growth, environmental pollution and negative globalisation-related symbolism. Yet, the relatively small size of the cruise sector and the over-focus on emissions arguably misrepresents the overall impact and potential of this tourism domain for portside communities, economies and ecosystems. Cruise-related scientific research, as probably expected, offers a much more refined and holistic picture, transcending the somewhat populist public debate on this matter. Based on a systematic literature review examining cruise-related papers published between 1983 and 2009, Papathanassis and Beckmann (2011) *Annals of Tourism Research* 38(1), 153–174, identified 145 papers, which were subsequently subjected to a metadata- and a thematic-analysis. Approximately, a quarter of them addressed the environmental-, social- and economic impacts of cruising on coastal regions. A decade later, and following an analogous methodological approach, a total of 305 cruise research papers, published between 2012 and 2022, yielded 161 relevant papers, subjected to the same coding scheme and thematically compared to previous findings. The subsequent thematic analysis, revealed a comprehensive set of issues, opportunities and challenges cruise tourism poses to coastal areas. Following a critical discussion of past developments and their trajectory, a future research and action agenda is proposed.

## Impact statement

Over the past decades and with the onset of the COVID-19 pandemic, the cruise sector has repeatedly been at the centre stage of popular discourse, featuring as a prime example for unsustainable tourism. The lion's share of the ongoing debate regarding cruise tourism sustainability revolves around ship emissions and waste disposal, raising questions on improvement feasibility and even on the desirability of cruise tourism altogether. As pertinent and valid this discussion may be, it implicitly limits the scope of cruise tourism activity and accountability within the physical and organisational boundaries of cruise vessels. When it comes sustainability questions, such a ship-bound and operator-centric spotlight, back-stages the role and significance of an array of (coastal) destination stakeholders. Moreover, limiting focus and corrective action to a single sustainability dimension (i.e., environmental) and part of its aspects (e.g., emissions and waste disposal), carries the risk of spill-over effects on (and from) the economic and socio-cultural dimensions. Conducting a comprehensive review of the scientific research literature published over the past decade manifests a more comprehensive picture. From the selected 305 cruise-related papers, published between 2012 and 2022, 161 (i.e., 50.7% of the total) addressed one or more sustainability dimensions. Apart from underlining the centrality of sustainability, the research community echoes a more holistic and systemic approach to cruise tourism, addressing the multi-dimensionality of sustainability and transcending the polarisation and oversimplification observed in the press and popular debate. Aside from the proportion of scientific literature focusing on mitigating the negative environmental impacts of cruise tourism (33%), a substantial proportion of papers address the challenges posed by the economic dimension of sustainability (42%) and the potential/possibilities of better managing the environment-economy trade-off (27%). The themes emerging from the reviewed scientific literature provide a comprehensive synopsis of relevant research areas and domains of effective sustainable action.

## Introduction and background

### The 'anti-hero journey' of cruise tourism: A tale of innovation, growth, resilience and reckless capitalism

Up to the outbreak of the COVID-19 pandemic, the narrative of cruise tourism was one of double-digit annual growth in terms of passengers, increasing ship sizes and technological innovations. Cruises, as a form of organised tourism, entered the mainstream in the 1980s as a reaction to the gradual displacement of passenger shipping by air travel, particularly for transatlantic crossings. Ship operators, faced with declining demand, enriched the maritime transportation core service with hospitality and entertainment activities on board. Over time, cruise ships evolved from hotels at sea to floating resorts. From the 1980s to 2018, the global cruise fleet grew from 79 to 369 vessels operating worldwide (Papathanassis, 2019a). A corresponding growth occurred with respect to average cruise ship size and capacity, from 19.000 to 60.000 gross registered tonnage (GRT) (Papathanassis, 2019a). A modern mega-ship, like Royal Caribbean's Symphony of the Seas (~228.000 GRT) is approximately five times the size of the Titanic (~46.000 GRT) (Papathanassis, 2019a).

At its core, a cruise holiday combines in a single package: the experiential richness of a roundtrip, the comfort and amenities of a hotel, and the entertainment options of a holiday resort. The cost-effectiveness of larger cruise vessels, as well as an array of technological and service-related innovations on-board, have rendered cruises more affordable and comparable to a high-end land-based holiday in terms of end price (Papathanassis, 2022, p. 688). The popularity and democratisation of cruise holidays over the past decades can also be attributed to a modernisation of their image, appealing to a wider spectrum of customer segments, mutating cruising from a holiday niche activity to a mainstream holiday form. At its peak in 2019, the cruise sector counted 27.5 million passengers, 179.7 million (cruise) bed-days, 1.17 million full-time equivalents (FTEs), and $150 billion in direct, indirect and induced economic benefits (Papathanassis, 2022, p. 689).

The growth and popularity of cruise tourism were inevitably accompanied by increased visibility in the public domain. The narrative of continuous growth and success was welcomed and amplified by the sector as a means of attracting first-time cruisers and increasing market penetration. Public scrutiny and censure, particularly on sustainability and corporate social responsibility (CSR) issues, proved to be the flipside of free publicity. Critics, including environmental groups (e.g., 'Friends of the Earth'), online communities (e.g., International Cruise Victims Inc.), various academics and experts (e.g., Klein, 2010, 2011; Klein, 2022), and even local communities (e.g., anti-cruise ship demonstrations in Venice or Barcelona) have been consistently pointing at the cruise business' failings and malpractices in each and every sustainability dimension.

The public relations approach adopted by the industry's representational entity Cruise Lines International Association (CLIA) to counter negative publicity has been a rather defensive one, attempting to redirect attention to the positive economic impacts and technological developments of the industry. In addition, the sector repeatedly claims their role as early adopter and leader of sustainability practices in the wider shipping sector, whilst (and despite the fact) they represent only a minuscule fraction of it. Medially, the cruise business and its reaction to sustainability- and CSR-related reproaches, convey an archetypical 'anti-hero journey' storyline for journalists, bloggers and the media. The narrative of a resilient, successful growth industry 'breaking bad' is arguably a basis for

engaging and ongoing news coverage. Below are some examples of headlines over the past years:

- Guardian (09/07/2022): 'U.S. cruise ships using Canada as a "toilet bowl" for polluted waste'.[1]
- Associated Press (14/07/2021): 'UNESCO: Italy's ban on cruise ships in Venice is "good news"'.[2]
- Financial Times (05/06/2019): 'Carnival cruise ships more polluting than all of Europe's cars'.[3]
- Deutsche Welle (21/06/2019): 'Are cruise ships climate killers?'.[4]
- Forbes (26/04/2019): 'Cruise Ship Pollution Is Causing Serious Health And Environmental Problems'.[5]
- The New York Times (05/01/2015): 'Cruise Ships Are Unregulated Trouble on the High Seas'.[6]

The onset of the COVID-19 pandemic saw the cruise business featured in the headlines with the Diamond Princess incident, and the subsequent characterisation of cruise ships as 'floating Petri dishes'. The pandemic resulted in a drop of 74% in passengers, carrying around 7 million in 2020 and slightly recovering to a total of 14 million in 2021 (Papathanassis, 2021). During the 2-year period several, mainly older, vessels were decommissioned and/or dismantled, as to reduce the operational costs required to maintain the docked cruise fleets. For the 'big 3' cruise operators (i.e., Carnival Cruise Lines, Royal Caribbean Cruise Lines and Norwegian Cruise Lines), the 'no sail cash-burn' amounted to some $1 billion per month (Staff, 2020). Effective cost management and the ability to keep afloat with the help of $12 billion of new debt and own equity capital, allowed them to 'stay afloat' and gradually resume operations in 2022. The global cruise fleet, has slightly shrunk (~300 vessels in total), with an order book of an additional 107 vessels expected to come into operation by 2027 (Staff, 2021).

### Cruise sustainability myopia: Cruise ship-centrism

As the cruise industry resumes its growth trajectory, demonstrating its financial resilience and market demand, the public sustainability debate has re-emerged, with cruise operators committing "make zero-emission vessels and fuels widespread by 2030, and to achieve a goal of 'net-zero carbon' cruising by 2050" (Palmer and Palmer, 2022). Besides improvements in propulsion and energy management on board, measures to reduce emissions involve the use of alternative, cleaner fuels (LNG), shore-side electricity and scrubbers. Whilst the public debate is focusing on the effectiveness and feasibility of various 'green technologies' and sustainability measures (Jainchill, 2021; Barnes, 2022; Syal, 2022), its scope and depth remain fairly limited. More specifically, the current discussion fails to adequately consider:

---

[1] https://www.theguardian.com/environment/2022/jul/09/us-cruise-ships-using-canada-as-a-toilet-bowl-for-polluted-waste-alaska-british-columbia.

[2] https://apnews.com/article/europe-business-health-government-and-politics-coronavirus-pandemic-9dd995a6a0d6d249e1e0e49b143f5408.

[3] https://www.ft.com/content/8bceef94-86cd-11e9-a028-86cea8523dc2.

[4] https://www.dw.com/en/are-cruise-ships-climate-killers/a-49275044.

[5] https://www.forbes.com/sites/jamessellsmoor/2019/04/26/cruise-ship-pollution-is-causing-serious-health-and-environmental-problems/?sh=b0d2d4737db3.

[6] https://www.nytimes.com/roomfordebate/2014/12/29/how-tourists-can-do-less-harm-than-good/cruise-ships-are-unregulated-trouble-on-the-high-seas.

- Cruise guests' full holiday journey (e.g., pre-embarkation and post-disembarkation transportation, pre- and post-cruise programme, port-excursions).
- Cruise ships' entire life cycle (i.e., design and construction, operation and maintenance, dismantling and recycling).
- Other sustainability dimensions (i.e., environmental, social, cultural and economic).

Indeed, the larger part of cruise holidays takes place on shore, a cruise vessel has a limited operational lifespan and the impacts of cruising activity on coastal areas extend beyond emissions and waste disposal. A systemic view on cruise tourism, in line with the definition proposed by Papathanassis and Beckmann (2011), calls for a more differentiated and wide-encompassing discourse of cruise sustainability:

> Cruise tourism is a socio-economic system generated by the interaction between human, organisational and geographical entities, aimed at producing maritime-transportation-enabled leisure experiences. (p. 166)

Therefore, the purpose and aim of this review are twofold: first, to provide an overview on the development of cruise tourism research over the past decade and evaluate the thematic importance of sustainability in this context; second, to examine the scientific community's outlook on the corresponding discussion, and its potential for contributing to the ongoing public debate.

## A decade of cruise research: Sustainability at the focus of academic discourse

Papathanassis and Beckmann (2011) conducted a comprehensive literature review on cruise tourism-related research published between 1983 and 2009. Systematically mining mainstream bibliographic databases, the authors conducted a descriptive and thematic analysis of 145 cruise-related academic publications. The sustainability theme addressed the environmental, economic and social impacts of cruising on coastal areas and reflected a quarter of the scientific literature coded. The dominant theme identified by the authors was labelled 'cruise market' and reflected the 'managerialism-focus' of research and practice at the time, namely, a taken-for-granted growth paradigm and the challenge of exploiting its market potential.

> The first theme (Cruise market') focuses on the relationship between cruise operators and their actual and potential passengers. Approximately one third of cruise-related papers are primarily concerned with marketing practices in the cruise sector. Market trends and consumption analysis, demand forecasting, customer expectations and pricing issues remain highly relevant; presumably reflecting their significance for the sector and its business stakeholders. (Papathanassis and Beckmann, 2011, p. 164)

Following an analogous approach to that of Papathanassis and Beckmann (2011), over a decade later in May 2022, the keywords 'Cruise' and 'Tourism' were used to query the databases of Elsevier (https://www.sciencedirect.com/search), Taylor & Francis (https://www.tandfonline.com/) and Emerald (https://www.emerald.com/insight/). The search was refined by filtering for review and research articles (ARTICLE TYPE) published between 2012 and 2022 (PUBLICATION YEAR). From the resulting bibliographic sample of 317 articles and upon closer examination, 12 were excluded from the sample due to lack of thematic relevance (i.e., not addressing cruise tourism), and 161 were thematically focused on sustainability issues pertaining to cruise tourism, reflecting almost half of the

sampled scientific literature, published between 2012 and 2022. More specifically, 70 of those papers focused on the environmental sustainability dimension, whilst 91 on the economic and social aspects of cruising on coastal regions and their communities.

Merely comparing the proportion of scientific papers addressing cruise tourism sustainability between the past two decades, it becomes apparent that cruise sustainably has become a dominant theme within the academic community (see also Wondirad, 2019), mirroring the focal attention of public opinion, the press, and the sector's critics and advocates. The main relevant question here is the degree of thematic diversity covered by the scientific debate and the extent to which it transcends the mono-dimensional emission-focused, ship-centric debate in the public domain.

## Deconstructing cruise tourism sustainability: Thematic analysis and discussion

The themes identified through the systematic analysis of the selected papers could be synthesised in the form of answering the following metaphorical, and perhaps even partly literal 'million dollar question':

> What are the conditions, activities and measures required to effectively balance the trade-off between maximising economic potential and minimising the corresponding sustainability-related risks of cruise tourism for coastal areas?

Conceptually, the thematic analysis is depicted in Figure 1.

### *Exploiting the economic potential of cruise tourism: Destination-competitiveness, -management and post-COVID-19 reality check*

The academic community and the corresponding research offer a fundamentally different, more systemic, narrative than the mainstream media. The casual and uninvolved media consumer, may be tempted to unilaterally ascribe cruise tourism an image, of dominant corporations practically imposing their unsustainable business models and exploiting coastal destinations' natural and cultural resources and they are harming their communities. Yet, cruise tourism can also be beneficial in a number of ways, not mentioning that the actual power dynamics within the cruise supply chain are considerably more complex, multi-levelled, and involve a variety of stakeholders and decision-makers (Papathanassis and Bundă, 2016, p. 165). Whilst the aspect of (negotiation) power in the cruise supply chain has been addressed in the past (Pranić et al., 2013; van Bets et al., 2017; Puriri and McIntosh, 2019), it has inevitably resurfaced following the pandemic-induced sail-stop (James et al., 2020; Renaud, 2020; Alberini, 2021; London et al., 2021; McLean, 2022; Spencer and Spencer, 2022).

Particularly for established/mainstream cruise destinations and regions (i.e., Caribbean, West Mediterranean), the sail-stop during the pandemic years and the corresponding collapse of (cruise-) tourism income and employment, has led to a reality-based re-assessment of economic dependence and resilience, for both the cruise-operators and for coastal destinations. Cruise operators such as TUI Cruises experimented with offering 'Cruises to Nowhere', which involved only days at sea (Steinbuch, 2020), whilst established destinations such as Venice consecutively experienced void tourism seasons (Nast, 2021).

On this basis and with the gradual recommencement of cruising during 2022, the realignment of power balances calls for an adjustment of destination strategies and policies (Franklyn-Green et al.,

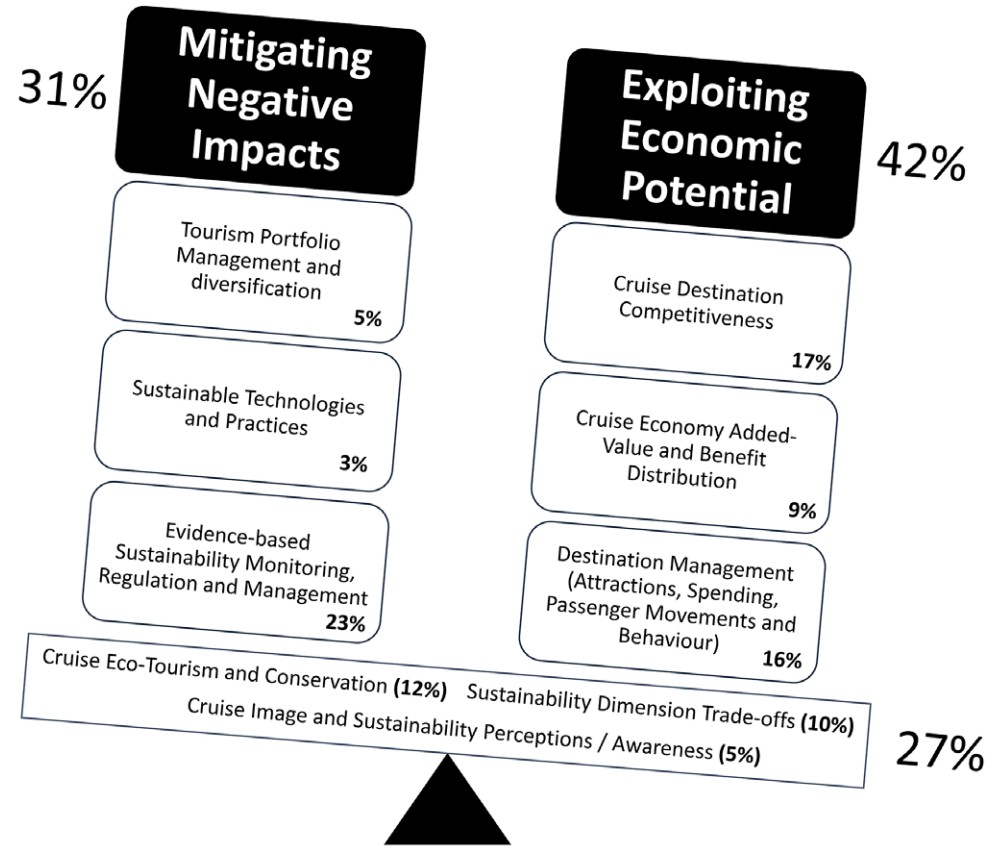

The percentages reflect the proportion of published sources in the selected sample for this systematic literature review (N=161)

**Figure 1.** Thematic analysis – Cruise-related published research 2012–2022 (*N* = 161).

2022). Central to this discourse is the maximisation of cruise tourism's value-added for destinations; destination competitiveness being the primary driver of doing so.

### Cruise economy added-value and benefit distribution: The precondition of institutional effectiveness

The question of cruise tourism's economic value-added entails fore and foremost the financing models (Sugimura, 2020) and investment returns for cruise- and tourism-related infrastructure, logistics and business services (Urbanyi-Popiołek, 2014; Artal-Tur et al., 2019), especially in the context of the post-pandemic era (Ajagunna and Casanova, 2022). A more equitable sharing of investment costs and risks could be achieved through private–public partnerships (PPP) (Sugimura, 2020) and/or through regulating ownership and exercising control over tourist venues and attractions (Puriri and McIntosh, 2019). Besides the direct economic impacts related to cruise tourism, there is also an array of indirect costs involving the local residents' quality of life (Jordan and Vogt, 2017), the availability and access to food and water (MacNeill and Wozniak, 2018), as well as their psychological well-being (McFarlane-Morris, 2021). Whilst estimating and monitoring direct, indirect and induced economic impacts represents a methodological challenge (Artal-Tur et al., 2019) at the destination level, facilitating a more equitable distribution between a local community's social groups poses an additional challenge for policy-makers (Del Chiappa et al., 2018).

Taking all the above into account and considering the inconclusive and conditional nature of cruise tourism's impact on coastal regions and communities, one can deduce that the presence of an appropriate regulatory architecture and the ability to uphold and control it is a key precondition. This, in turn, dictates institutional effectiveness and public authority competence.

### Cruise destination competitiveness: A question of geography, infrastructure and regulatory framework

Destination competitiveness reflects the strategic dimension of cruise tourism's economic development and aims at maximising demand by improving the source-market attractiveness and regional positioning of a port destination.

Home-porting (i.e., port of embarkation and disembarkation) is considered more favourable than serving as a port-of-call within an itinerary, as it is usually associated with higher cruise B2B-related income and visitor spending (Jordan, 2013; Bagis and Dooms, 2014; London and Lohmann, 2014; Niavis and Vaggelas, 2016; Chen et al., 2019). This can be mainly attributed to longer lengths of stay at port (Chen and Nijkamp, 2018), business and logistics services to cruise operators (Chen, 2016), as well as revenue and employment generated via guest-transportation services and facilities, pre- and post-cruise overnight stays.

The main determinants of (port) destination competitiveness can be grouped under the following headings:

- *Geography*: The geographical position within a given cruise itinerary, as well as the corresponding connectivity of ports with other destinations and their hinterlands (Tsiotas et al., 2018; Tseng and Yip, 2021; Yuksel et al., 2021).
- *Infrastructure*: The functionality and capacity of the available infrastructure (Sun et al., 2014; Lopes and Dredge, 2018), along

with the general degree of economic and tourism development in the area (London and Lohmann, 2014; Niavis and Vaggelas, 2016; da Luz et al., 2022).

- *Regulatory framework*: Dawson et al. (2017) adopt the example of the Canadian Arctic to highlight the negative effects of (over-)regulation and bureaucracy on destination attractiveness, and make a case for improved process- and organisational efficiency in the implementation of regulations.

The existence of a favourable institutional architecture, combined with a suitable strategy for improving competitiveness, is alone no guarantee for cruise-driven economic success unless the strategy is effectively managed and translated at the operational level.

### Destination management: Professionalism imperative for attraction protection and visitor spending

The sustainable management of attractions (McLean, 2022), critical tourism resources (Sun et al., 2021) and the protection of heritage sites (Marsh, 2012; Colavitti and Usai, 2019) require seasonality control, crowd management, stakeholder coordination and business regulation (Castillo-Manzano et al., 2015). At the operational level, understanding and managing cruise-visitor movements in destination areas and attractions, their behaviours, spending-patterns and interaction-valence with the local community have been extensively studied (Weaver and Lawton, 2017; Ferrante et al., 2018; Brandajs and Russo, 2021; Casado-Díaz et al., 2021), often in the context of the overtourism debate (Abbasian et al., 2020; Shoval et al., 2020; Baumann, 2021; Ilori et al., 2022). From this perspective, overtourism can be seen as the result of destination-operational and managerial failure.

Direct spending, word-of-mouth marketing and intention to return are the main value-contribution of cruise visitors to destinations (Parola et al., 2014; Penco and Di Vaio, 2014; Sanz Blas and Carvajal-Trujillo, 2014). The main driver of cruise visitor's spending and their intentions to return is (not surprisingly) guest satisfaction, stemming from authentic, positive experiences and service interaction quality (Baumann, 2021; Kim et al., 2021). Other determinants of cruise destination visitor-satisfaction extracted from the literature include local prices (Brida et al., 2015), length of stay (Sanz-Blas et al., 2019; Casado-Díaz et al., 2021), visitor demographics (López-Marfil et al., 2021), destination image (Toudert and Bringas-Rábago, 2016), visit-excursion arrangements (Navarro-Ruiz et al., 2020), travelling/excursion group size (Brida et al., 2012) and even weather conditions (Baños-Pino et al., 2022).

The multiplicity of institutional, regulatory, strategic and managerial challenges mentioned previously is evidently also moderated by a multi-variate set of behavioural patterns exhibited by the various stakeholders involved. This underlines the need for a systemic view of the cruise phenomenon. A simplistic focus on ship-call counts and cruise visitor volumes by tourism policy-makers and by the press, grossly fails to capture the development dynamics and conditional nature of cruise tourism's economc potential. In the absence of a visible, or perceived for that matter, economic justification for cruise tourism, its negative impacts and risks for the environment and other socio-economic activities feature prominently in the public perception and political attention.

### Mitigating the negative impacts of cruise tourism: Eco-diversity, regulation and best practices

The environmental impact of cruise tourism has consistently received attention from both the press and the scientific community. Specifically, reducing the emissions and waste originating from cruise ships remains a challenge in regulatory as well as technological terms.

### Evidence-based sustainability monitoring, regulation and management: 'Measure it to control it!'

The measurement and quantification of cruise environmental impacts on coastal regions is an integral part of monitoring and controlling them. A proportion of research has invested in comparative quantitative analysis between regions and/or other forms of tourism/economic activity (Gössling et al., 2015, 2017; Lin et al., 2018; Lloret et al., 2021), underlining the detrimental environmental impacts of cruising and advocating its stricter regulation as a less preferable form of tourism.

In terms of measuring and monitoring emissions, researchers are faced with the difficulty of modelling and isolating the proportional impact of cruise ships in particular, and cruise tourism in general (Simonsen et al., 2019; Stewart et al., 2020). For example, Pagoni and Psaraki (2014) analysed and quantified the aircraft emissions produced by the aircraft system and assessed the impact of tourism demand volatility on air pollution. Departing from a ship-centric view of cruises, and taking into account the entire cruise-passenger journey (e.g., air and land transportation), restricting emission measurement to port areas does not provide a full picture (Gössling et al., 2015, 2017; Grythe and Lopez-Aparicio, 2021). Then there is the problem of isolating the proportion of emissions produced by different forms of shipping at a port. Here, comparing the level of emissions between peak and off-peak tourism seasons, to subsequently attribute the difference to cruise ships (Papaefthimiou et al., 2016) is also questionable.

Moreover, a case can be argued against monitoring and controlling emissions as an isolated measure for managing the environmental impacts of cruise tourism. Various measurement/indicator improvements, along with existing model-/index-extensions, have been proposed to better address other aspects of cruise-induced environmental impacts (e.g., Vicente-Cera et al., 2020). Others have argued for a wider scope, in conjunction with an incorporation of other sustainability dimensions (Carić and Mackelworth, 2014; Carić, 2016; Chen and Chen, 2016; Pesce et al., 2018; Wang et al., 2020) and CSR in general (Papathanassis, 2017; Lin et al., 2018; Nikčević, 2019; Ramoa et al., 2020).

Indeed, at the heart of cruises' environmental impact lies the issue of understanding and measuring its magnitude in order to control and regulate it. A further question in this context refers to the actual capacity to do so. The sustainable management of tourism and general and cruise tourism in particular is a complex managerial undertaking, requiring the involvement of multiple stakeholder groups, at both the local and regional levels (Cajaiba-Santana et al., 2020). Aligning a network/cluster of diverse perceptions and competing private and public interests, whilst ensuring acceptable levels of community participation, is at the core of successful policy implementation (Viken and Aarsaether, 2013; Garay et al., 2014; Harkison and Barðadóttir, 2019; Dimitrovski et al., 2021).

### Tourism portfolio management and diversification: 'Tourism-monocultures' are not sustainable

Juxtaposing the environmental and social risks of cruise tourism against the economic livelihood dependence of some communities, renders the aspiration to regulate and control environmental risks somewhat of a theoretical exercise. This is the point where the

power balances and economic dependencies in the cruise supply chain, mentioned before, supress the readiness to mitigate and control sustainability-related risks. Over-dependence on a particular tourism form and/or segment is the equivalent of an 'economic monoculture', which is arguably a risky and crisis-prone option as it is subject to seasonality, economic cycles and corporate exploitation (James et al., 2020; Renaud, 2020). Diversification and a balanced tourism portfolio enable improved capacity risk management (Jeevan et al., 2019; Jordan et al., 2020) and investment steering (Pulina et al., 2013).

Whilst the case for economic diversity and a seasonally-balanced tourism portfolio is straightforward and self-evident, particularly in the aftermath of the pandemic on tourism activity, the very persistence and emergence of tourism monocultures indicate its evasiveness in practice. The structural aspects of tourism system (i.e., international supply chains, predominance of small–medium enterprises, market-share concentration) subject it to corruption, which simultaneously echoes and disrupts economic-development and sustainability-related progress (Papathanassis, 2019b). Systemic corruption in the context of preventing sustainable tourism development (in all dimensions) can be attributed to institutional failure, and affects the ability of coastal regions to strategically manage their tourism activities and resources. Institutional effectiveness, as mentioned earlier, is a precondition for destination competitiveness, both of which constitute the framework for sustainable tourism development.

### Sustainable technologies and best practices: Underrated potential and adoption timing

Improvements in propulsion technology and energy efficiency have consistently played a major role in shipbuilding, particularly over the past years, due to stricter regulations and rising energy costs (Amaya-Vías et al., 2018; Eikeland et al., 2020). Technological innovations in shipbuilding, concerning waste-treatment and management also imply cost reductions (Paiano et al., 2020), whilst improving environmental sustainability (Carić et al., 2016). Technological innovation is often a domain where profit maximisation and ecology coincide. From the approximately 100 new builds planned up to 2027, one-fifth are LNG powered, corresponding to 39% of the new tonnage and 41% of the added capacity (Papathanassis, 2021, p. 17). This and a closer view of the current cruise ship order book reveals that LNG-powered vessels tend to be larger in terms of tonnage and passenger capacity and thus more expensive (+59%), in part due to their size. As older vessels are decommissioned and newer, more technologically advanced, vessels join the global cruise fleet, cleaner fuels and energy-saving technologies will gradually displace other, older and less-sustainable technologies. The reason, why the technological aspect has assumed a rather secondary role in scientific research and public debate is probably because it is in fact a delayed evolution, a reaction to the overdue revolution dictated by the zeitgeist-action-urgency of climate change.

### Transcending the 'economy-ecology dichotomy': Cruise citizen-science, conservation and awareness

Whilst technological innovation promises sustainability progress in the longer term, there are also direct and indirect contributions to sustainability, related to certain expressions and sub-types of cruise tourism.

### Cruise-eco tourism and conservation: Fostering responsibility
Besides the fairly intuitive popularised proposal of tourism tax for conservation purposes (Verbitsky, 2015; Sanches et al., 2020), Storrie

et al. (2018) and Taylor et al. (2020) highlight the potential of environmental citizen science by utilising data collected by cruise passengers, particularly those who participate in expedition cruises. Furthermore, in a number of regions, cruise tourism may be the 'least-worst-alternative' when it comes to sustainability. Dependence on eco-tourism may act as an economic/political counterweight (or moderator) against other, less sustainable economic activities such as oil drilling (Gould, 2017), dredging (Lester et al., 2016), electricity production (Paoli et al., 2017; Callejas-Jiménez et al., 2021) and fishing (Lasso and Dahles, 2018). Another indirect, but yet mention-worthy positive effect of expedition cruise tourism is its educational and awareness-creation potential for sustainability values and issues (Walker and Moscardo, 2014; Ardoin et al., 2016; Zander et al., 2016; Warmouth et al., 2021). Tourism can transform a 'sense of place' to a 'care of place' encouraging tourists and locals to assume more responsibility (Walker and Moscardo, 2016) and play a more active role with regards to conservation (Hillmer-Pegram, 2016).

### Sustainability dimension trade-offs: Beyond sensationalism and populism
Thus far, the literature reviewed for this paper makes it rather clear that the topic of sustainability and the impact of cruise tourism on coastal regions does not yield simple answers and necessitates a critical yet panoptic approach. Failure to consider the spill-over effects between the sustainability dimensions can result in misplaced actions and measures, addressing one problem whilst triggering several others (Robinson et al., 2019; Ren et al., 2021). A systemic understanding of sustainability and the trade-offs between its dimensions is a precondition for effectively managing it. In addition, there is a need to differentiate between the impacts and potential of various sub-segments of (cruise) tourism (Cheung et al., 2019; Wahnschafft and Wolter, 2022) and understand their interaction with the destination context (Tin et al., 2016; Panzer-Krause, 2020).

Despite, highlighting its relevance and inter-disciplinary potential, deriving and synthesising knowledge from the scientific literature, enables a multi-faceted view of sustainability in the (cruise) tourism context. This can reduce the risk of exploiting tourism and its significance for media-sensationalism (Nagel, 2020), and worse even for political and/or corporate interests, particularly for the developing economies of the global south.

### Cruise image and sustainability awareness: Slow but steady!
A casual view of the press coverage regarding tourism in general and cruises in particular constructs the picture of a few 'green-washing', exploitative cruise operators, serving hedonistic and environmentally insensitive customers. Despite cases and practices reinforcing this view, this is only part of the picture. For one, the 'business' of cruising entails a larger set of actors and stakeholders, ranging from cruise operators, industry associations and NGOs to local public authorities, businesses and independent small vendors. Indirectly, 'producing holiday experiences' involves entire communities and is subject to a multiplicity of motives, interests and perspectives. In other words, the cruise business does not merely consist of cruise corporations, but an entire network of institutional and business actors, who set the stage and participate in (un-)sustainable practices.

Secondly, the notion of 'hedonistic', environmentally-insensitive cruise tourists is also contestable. As eco-awareness increases amongst the population at large, sustainability considerations are playing an increasingly important role in the booking decisions of cruisers (Roura, 2012; Han et al., 2019), which in turn increases

pressures on the sector to review and adapt their business models and practices. A transformation driven by shifts in (cruise) tourist attitudes and booking behaviour, resembles a bottom-up, organic evolution. This may well be the key for effective and lasting change towards more sustainable forms of coastal tourism, but nonetheless a Sisyphean challenge with delayed results. Therefore, it needs to be complemented by top-down regulation at multiple levels, ranging from international to local.

## Final word and the coming decade

'Rethinking tourism' and focusing on positively transforming it 'for both people and planet', was the central theme of the 2022 World Tourism Day celebrated on the 27th of September.[7] Sustainable tourism is one of the main touchpoints of the UNWTO's[8] 'Rethinking tourism' campaign, and its premises are also readily transferable to cruise and coastal tourism:

> Sustainability principles refer to the environmental, economic, and socio-cultural aspects of tourism development, and a **suitable balance** must be established between these three dimensions to guarantee its long-term sustainability… Sustainable tourism development requires **the informed participation** of all relevant stakeholders, as well as **strong political leadership** to ensure wide participation and consensus building.[9]

The tourism academic community, and their research efforts over the past decade, have acknowledged the systemic complexity and multidimensionality of cruise tourism sustainability, whilst addressing the challenge of creating optimal equilibria in this context. Not only has the proportion of sustainability-focused cruise research practically doubled over the past decade, but also integrates a 'suitable balance' of all three sustainability dimensions. Nonetheless, the UNWTO's call for stakeholder information and wide participation as a footing for political will and purposeful action, goes beyond knowledge creation and calls for knowledge transfer. The undifferentiated and simplistic representations of cruise tourism often encountered in the media and the current, ship-centric, focus of regulatory action (or inaction for that matter), are indicative of a disconnect between scientific research and its practical relevance and impact. Dickhut and Tenger (2022) analysed the sustainability policies, strategies and instruments of 21 European countries and reported a "gap between good theoretical approaches and the general willingness to support a sustainable tourism development and the realisation of it" (p. 502), concluding that "in hardly any of the countries is sustainable tourism put in the centre of the national tourism policy as a priority area" (p. 501).

Our systematic literature analysis shows that 'cruise image and sustainability perceptions' (5% of the published sources sample), and 'sustainable technologies and best practices' (3% of the published sources sample) represent the least researched aspects of cruise tourism sustainability. Paradoxically, those very categories/themes are the very knowledge domains offering the highest potential for actionable problem-solving. Increased and informed public awareness enables the transition from media-sensationalism and political opportunism, to 'solutionism' and the collective political alignment required. In that vein, technology and innovation can facilitate and accelerate this transition.

For this reason, scientific communication is currently as relevant and challenging as ever, especially for a fragmented, pre-paradigmatic and multidisciplinary field such as tourism. Globally, the lion's share of tourism activity and impact takes place in coastal regions and is strongly interfaced with maritime transport and recreation. Its significance, both in terms of real and symbolic impact, renders it a key domain for sustainability action. Effectively disseminating and transferring the body of available cruise/tourism-related scientific knowledge outside the sphere of a small research community goes hand in hand with deciphering the determinants of public awareness, perceptions and behaviour, beyond the scope of the sector's main actors and direct beneficiaries. This, in turn, poses the challenge of synthesising a comprehensive body of relevant research and translating it into key messages and clear actionable responsibilities and activity domains. Although the scope of this review is limited to a handful of scientific databases, contains exclusively English-language publications and is thus by no means complete, it is arguably representative and comprehensive in terms of the issues and aspects it uncovers, summarises and synthesises. In this sense, we hope that this paper can serve as a starting point for relevant research in cruise tourism, adding an additional piece of the puzzle towards sustainable coastal futures.

**Open peer review.** To view the open peer review materials for this article, please visit http://doi.org/10.1017/cft.2023.2.

**Supplementary materials.** To view supplementary material for this article, please visit http://doi.org/10.1017/cft.2023.2.

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
