## [Reviewer Report]

*Comments to Author*: 1.Review papers need to have description of method with keywords used, search engine used etc ... to formthe list of primary literature and maybe other appropriate technical references. List of analyzed articles wouldhave to be enclosed in an appendix.

2. Since prof. Ross Klein was one of the most productive academics onthe topic his books should be noted.

3. There is no reference on the issues such as: crime on board, laborrights, not obeying the laws and regulations, etc.

4. The following sentences are opinions, lack references, areunclear in their meaning or logic (not all were listed due to lack of space): - Yet, the relatively small size of thecruise sector and the over-focus on emissions ultimately result to a misrepresentation of the overall impactand potential of this tourism domain for portside communities, economies and ecosystems. - From the 80s upto 2018, the global cruise fleet grew from 79 to 369 vessels operating worldwide. (reference) - e.g. anti-cruiseship demonstrations in Venice (there were many other cities and local communities demonstrating againstcruise tourism) - ecological (should be substituted with environmental) - The casual media consumer, issubjected to an unilateral image, depicting dominant (cruise) corporations practically imposing theirunsustainable business models and exploiting coastal destinations’ natural and cultural resources and theirharming their communities. (example of an opinion without proper proof or quote) - Yet, cruise tourism canalso be beneficial in a number of ways, not mentioning that the actual power dynamics within the cruisesupply chain are considerably more complex, multilevelled, and involve a variety of stakeholders anddecision-makers. (this is oppinion again, the relations are very simple and strait-forward: cruise company -port authorities - destination management ...) - Moreover, a case can be argued against motoring andcontrolling emissions as an isolated measure for managing the ecological impacts of cruise tourism (unclearmeaning of the sentence and the paragraph) - Indeed, at the heart of cruises’ ecological impact lies the issueof understanding and measuring its magnitude in order to control and regulate it. (this is not so, there is allotof research illustrating this, for example https://www.transportenvironment.org/discover/one-corporation-pollute-them-all/) - Technological innovations in ship-building, concerning waste-treatment and managementalso imply cost-reductions (Paiano et al., 2020), whilst improving ecological sustainability (Carić et al., 2016).(this is not what the authors of the cited article claim, quite opposite) - Technological innovation is often adomain where profit maximisation and ecology coincide. (this is not so for cruise industry as many studies onpoor quality of online waste treatment claim) - Despite, highlighting its relevance and inter-disciplinarypotential, diving into the scientific literature on the chosen review topic, can serve as an antidote againstexploiting tourism and its significance for media-sensationalism (Nagel, 2020), and worse even for politicaland / or corporate interests; particularly for the developing economies of the global south. (unclear sentence) -the paragraph Cruise image and sustainability awareness: Slow but steady! is poorly written and easilydisputable with available literature

5. The author is reflecting on many topics covered alluding on some sort ofthe research bias. However he/she neglects to note the underlying problem of lack of transparency of cruisecorporations that does not allow for the data to be researched.

---

## [Editor Report]

*Comments to Author*: One referee has provided substantial comments for you to consider. Please outline how you have responded to their comments in your revision in a separate reply to reviewers file with your submission.

---

## [Editor Report]

*Comments to Author*: Thank you for the very clear document outlining your response and revisions associated with each of the reviewer comments and questions. This is most appreciated by both reviewers and editors.